# GamePad: A Learning Environment for Theorem Proving

**Daniel Huang**[*][†]
dehuang@berkeley.edu

**Prafulla Dhariwal**[*][‡]
prafulla@openai.com

**Dawn Song**[†]
dawnsong@cs.berkeley.edu

**Ilya Sutskever**[‡]
ilyasu@openai.com

## Abstract

In this paper, we introduce a system called *GamePad* that can be used to explore the application of machine learning methods to theorem proving in the Coq proof assistant. Interactive theorem provers such as Coq enable users to construct machine-checkable proofs in a step-by-step manner. Hence, they provide an opportunity to explore theorem proving with human supervision. We use *GamePad* to synthesize proofs for a simple algebraic rewrite problem and train baseline models for a formalization of the Feit-Thompson theorem. We address *position evaluation* (*i.e.*, predict the number of proof steps left) and *tactic prediction* (*i.e.*, predict the next proof step) tasks, which arise naturally in tactic-based theorem proving.

## 1 Introduction

*Theorem proving* is a challenging AI task that involves symbolic reasoning (*e.g.*, SMT solvers (De Moura & Bjørner, 2008)) and intuition guided search. Recent work (Irving et al., 2016; Loos et al., 2017; Kaliszyk et al., 2017) has shown the promise of applying deep learning techniques in this domain, primarily on tasks useful for automated theorem provers (*e.g.*, premise selection) which operate with little to no human supervision. In this work, we aim to move closer to learning on proofs constructed with human supervision.

We look at theorem proving in the realm of *formal proofs*. A formal proof is systematically derived in a *formal system*, which makes it possible to *algorithmically* (*i.e.*, with a computer) check these proofs for correctness. Thus, formal proofs provide perfect learning signal—theorem statements and proofs are unambiguous. Human mathematicians usually do not write proofs in this style, and instead, construct and communicate proofs in natural language. Although the form and level of detail involved in each kind of proof differ, the logical content is similar in both contexts.

Our work focuses on *interactive theorem provers* (ITPs), which are software tools that enable human users to construct formal proofs. ITPs have at least two features that make them compelling environments for exploring the application of learning techniques to theorem proving. First and foremost, ITPs provide full-fledged *programmable environments*. Consequently, any machine learning infrastructure built for an ITP can be reused across any problem domain crafted to study an aspect of learning and theorem proving. Second, the proofs are constructed by humans, and thus, have the constraint that they must be relatively human-understandable. Hence, ITPs provide access to large amounts of supervised data (*i.e.*, expert-constructed proofs of theorems that are mathematically interesting). For example, ITPs have been used to build and check the proofs of large mathematical theorems such as the Feit-Thompson theorem (Gonthier et al., 2013) and provide provable guarantees on complex pieces of software such as the CompCert C compiler (Leroy et al., 2012).

We introduce a system called GamePad[1] that exposes parts of the Coq ITP to enable machine learning tasks and explore a few use cases. We focus on the Coq proof assistant for two reasons. First,

---

[*]Propositionally equal contribution
[†]University of California, Berkeley
[‡]OpenAI
[1]Theorem proving as a "game" and "proofs-as-data".

```
Lemma plus_O_nop:
  forall n: nat,
    n + O = n.
Proof.
  induction n; simpl.
  (* n = 0 *)
  reflexivity.
  (* n = n + 1 *)
  rewrite IHn.
  reflexivity.
Qed.
```

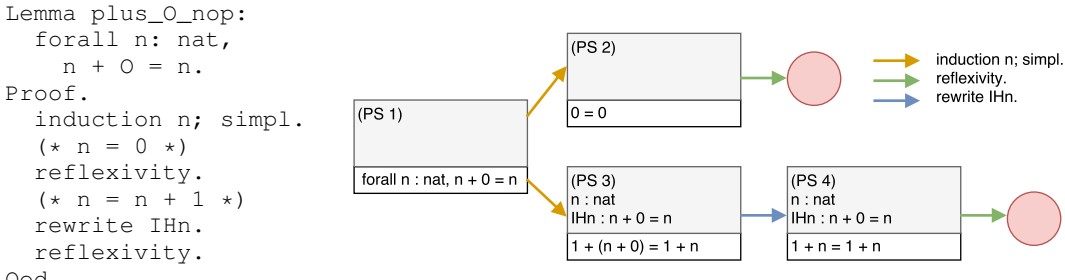

Figure 1: A proof script in Coq (left) and the resulting proof states, proof steps, and the complete proof tree (right). A proof state consists of a context (pink rectangles) and a goal (white rectangles). The initial proof state has as its goal the statement we are trying to prove and an empty context. The arrows indicate what tactic the prover used. The final states of the proof are indicated by the red circles and can be transitioned to only when the goal in the previous state is trivially true.

Coq is a mature system with an active developer community that has been used to formalize non-trivial theorems, including Feit-Thompson and CompCert. Second, Coq supports the *extraction* of verified software. Consequently, one can prove that a program is correct and then *run* the verified program. The ease of extraction makes Coq a popular choice for program verification.

Our contributions are the following. First, we introduce GamePad, which provides a structured Python representation of Coq proofs (Section 3), including all the proof states encountered in a proof, the steps taken, and expression abstract syntax trees (ASTs). The tool also enables lightweight interaction with Coq so that it can be used to dynamically build proofs (*e.g.*, used as an environment for reinforcement learning). Tasks that can leverage this structured representation (Section 4) include *position evaluation* (*i.e.*, predict the number of proof steps left) and *tactic prediction* (*i.e.*, predict the next proof step to take). We also discuss how we can use the structured representation to embed proof states into $\mathbb{R}^D$ (Section 5). Second, we demonstrate the synthesis of Coq proof scripts that makes use of a tactic prediction model for a hand-crafted algebraic rewriting problem (Section 6). Third and finally, we apply baseline models for position evaluation and tactic prediction to the Feit-Thompson formalization using data extracted by GamePad (Section 7).

The code for GamePad, as well as the associated data sets, models and results, are open source on GitHub at `https://github.com/ml4tp/gamepad`.

## 2 BACKGROUND

To provide context for the rest of this paper, we begin by illustrating the formal proof process and how it can be modeled as a game. We then walk through a simple example of a proof in Coq and end by summarizing the constructs that we will encounter in the rest of the paper.

**Theorem proving process** When humans write a pencil-paper proof, we typically maintain a mental and/or written *scratchpad* that keeps track of (1) what we need to show and (2) the facts we currently have at our disposal. As the proof progresses, the items in this scratchpad changes. For instance, we might transform what we need to show (*e.g.*, it suffices to show another statement) and/or discover new facts (*e.g.*, we derive an additional fact as a consequence of known facts). When we move from the pencil-paper setting to an ITP such as Coq, we will still have such a scratchpad, but use the ITP's *term language* to express mathematical statements and the rules of the ITP's *formal system* to construct the proof instead.

**Formal theorem proving as a game** A *game* serves as useful mental model of the proving process. The state of the game is a *proof state*, which looks like the scratchpad and is comprised of (1) a *goal* (expressed in the term language) stating what we need to prove and (2) a *context* containing the assumptions (also expressed in the term language). The *starting state* has as its goal the statement of a theorem and an empty context, while a *final state* has as its goal a statement that is trivially

true (*i.e.*, definitionally equal) given the context. The aim of a prover is to transform the starting state into a collection of final states using only *logically valid transitions*, which can affect both the context and the goal. It is possible to obtain multiple final states because some transitions may split a goal into multiple subgoals. For a given (sub)goal, a successful proof corresponds to a *proof tree*, where the root is the start state and all the leaves are final states.

**An example Coq proof**    Consider showing that adding $0$ to any natural number $n$ is $n$ itself. A paper-pencil proof might proceed by induction on $n$, where we check that the result holds on the base case (*i.e.*, $n = 0$) and the inductive case (*i.e.*, $n = n + 1$). We can carry out a similar process in Coq, using a sequence of commands called *tactics* in Coq's tactic language *Ltac* that indicate what proofs steps to take. For instance, we can use the code `induction n; simpl.` to start a proof by induction (on n) in Coq. The expressions `induction` and `simpl` are *tactics* (of arity 1 and 0 respectively), the semicolon `;` sequences two tactics, and the period `.` signals the end of one proof step. The effect of the entire command (up to the period) is to run induction and then simplify all resulting proof states and is shown in Figure 1 via the green arrows, which connect proof states 1 with 2 and 1 with 3. After we perform the induction, we see that proof state 2 contains the base case and proof state 3 contains the inductive case. The inductive case has an *inductive hypothesis* in the context. Proofs in Coq are finished when the goal is trivially true given the context (*e.g.*, `0 = 0` given an empty context). The collection of proof states and tactic invocations forms a *proof tree*.

**Formal definitions**    Coq provides three equi-expressive term languages that encode logical formulas such as `forall n: nat, n + 0 = n`. They differ in the amount of *annotation* they support. Coq's user-facing term language *Gallina* supports type-inference and other forms of notational convenience (*e.g.*, syntactic sugar and extensible parsing). Coq's *mid-level* term language removes all forms of notational convenience. Finally, Coq's *kernel-level* term language instantiates all types and is the level at which proofs are checked. Thus, the mid-level and kernel-level languages are similar, with the exception that the mid-level term language has a notion of an *implicit argument*.

Implicit arguments occur in *application constructs*—$M\ M_1 \ldots M_n$ (apply function $M$ to arguments $M_1 \ldots M_n$) at the kernel-level versus $M\ M_1^{\iota_1} \ldots M_n^{\iota_n}$ at the mid-level, where $\iota_i$ marks implicit versus non-implicit arguments. Implicit arguments take care of the book-keeping aspects of a formal proof such as the types of all terms involved; a human prover would usually omit them in a paper-pencil proof. Hence, learning with mid-level terms without implicit arguments is closer to human-level proving, whereas learning with kernel-level terms is closer to machine-level proving.

Coq represents a *proof state PS* as a tuple $\langle \Gamma, M, n \rangle$ of a local context $\Gamma$, a goal term $M$, and a unique proof state identifier $n$. A *context* $\Gamma$ provides a list of identifiers and their types that expresses all the current assumptions. For instance, the proof state with identifier 3 has $\Gamma = $ `n : nat, IHn : n = n + 0`. A tactic invocation creates edges between the appropriate proof states, which forms a *proof tree*.

Proof states are interpreted in the presence of *global state*. The global state contains the interpretation of constructs such as constants and constructors that have global scope. It also contains the interpretation of constructs that have stateful effects such as existentials. In particular, it is possible to posit the existence of an object in one proof state and then proceed to perform case analysis resulting in multiple proof states that share the same existential. The GamePad tool (Section 3) exposes the constructs described above (with the exception of Gallina terms) as Python data structures, which we can use for building models.

## 3    GAMEPAD TOOL

We briefly describe the GamePad tool with an emphasis on the *data* that it exposes and any other design decisions that affect the modeling process.

**Obtaining proof traces**    We obtain the proof trace by implementing a patch to Coq (version 8.6.1) that instruments the Coq Ltac interpreter to log the intermediate proof states. The patch also supports SSreflect (Gonthier & Stéphane Le, 2009), a popular Coq plugin for writing proofs about mathematics. Importantly, the patch does not touch the Coq proof checker, and hence, does not affect the critical part of the system that verifies the correctness of proofs.

This implementation choice affords us flexibility in deciding what proof steps to consider *atomic*. As a reminder, the sequenced tactic `induction n; simpl.` (Section 2) can be considered as a single proof step (read up to the delimiting `.`) as in the example, but it is comprised of two primitive tactics—`induction` and `simpl`. As the granularity of a proof step has consequences for setting up a tactic prediction task (Section 4), we made the choice to obtain the finer-grained proof states by modifying Coq directly while maintaining the mapping to the human-level proof script. Thus, the tool records proof states at the granularity of primitive tactics (*i.e.*, breaks up `;`) as well as at the granularity of the proof script (*i.e.*, `.`)

**Representing Coq proofs**   We expose the proof trace obtained from Coq as Python data structures, including Coq's mid-level and kernel-level term languages, proof states, proof steps, and proof trees so that they can be manipulated with arbitrary Python code. Consequently, we can build models using this structure. For efficiency reasons, we represent Coq terms in a shared form. Representing terms in a shared form is a necessary technique to scale compilers and ITPs to real-world programs and proofs. In our setting, it is also essential to scaling training/inference to real-world data sets (see Section 7).

**Light-weight interaction**   The API provides a thin wrapper around `coqtop`, the Coq repl, which can be used to interactively construct proof scripts from within Python (see Section 6). This component mediates all interaction with Coq, including proof state parsing into GamePad's representation of Coq proofs and sending tactic actions.

## 4   TASKS

In the theorem proving process, we encounter two important tasks: *position evaluation* and *tactic prediction*. A position evaluator $\mathcal{P}$ tells how easy it is to prove a given proof state, while a tactic predictor $\mathcal{T}$ tells how to take an action that can make our proof state easier to prove. As a reminder, proof states are interpreted in the presence of global state, which we have left implicit here. Note that in general, $\mathcal{P}$ depends on $\mathcal{T}$; a good tactic predictor would find it easier to prove a given proof state. Also, an action taken by $\mathcal{T}$ can lead to multiple child proof states, and thus $\mathcal{P}$ must consider the provability of all child proof states.

In this work, we aim to learn parametric functions $\mathcal{P}^{\mathcal{T}_h}$ and $\mathcal{T}_h$ using supervised learning on a data set of human proofs, where $\mathcal{T}_h$ is the human tactic predictor implicit in the data set. In the future, one can use $\mathcal{P}$ and $\mathcal{T}$ within an end-to-end prover, and learn them directly using reinforcement learning. This requires the ability to manipulate proof states by sending actions, which is possible using the light-weight interaction provided by GamePad.

**Position evaluation**   The goal of the *position evaluation task* is to predict the approximate number of steps required to finish a proof given an input proof state. We define this as the function $\mathcal{P}^{\mathcal{T}}$ : $PS \rightarrow \{1, \ldots, K\}$ for a given tactic predictor $\mathcal{T}$, where we bin the sizes of the proof trees into $K$ classes to make learning easy. Given a data set of steps taken by a human prover, we aim to learn $\mathcal{P}^{\mathcal{T}_h}$ by supervised learning on $N$ training tuples $\{(s_n, d_n)\}_{1 \le n \le N}$, where each $s_n$ is a proof state and each $d_n$ is the binned size of the proof tree below the corresponding $s_n$.

The position evaluator defined above provides a proxy for how difficult it is to complete the proof from the current state—a lower number indicates that the goal is easy to deduce given the context. A model for position evaluation can be used to create a sequence of tasks for curriculum learning or by human provers to measure the progress they are making in a proof. We also note that position evaluation contains aspects of the premise selection problem (Irving et al., 2016) in that it should assign a high number to proof states which do not yet contain the requisite hypotheses to prove the current goal.

**Tactic prediction**   The goal of the *tactic prediction task* is to predict the next tactic to apply given an input proof state. We define this as the function $\mathcal{T} : PS \rightarrow T$, where $T$ represents the set of possible tactic actions. Given a data set of steps taken by a human prover, we aim to learn the human tactic predictor $\mathcal{T}_h$ given $N$ training tuples $\{(s_n, t_n)\}_{1 \le n \le N}$, where $t_n$ is the tactic the human prover used in state $s_n$.

Tactic prediction is more localized to a single proof state compared to position evaluation, although there are two additional challenges. First, we must choose the *granularity* of a proof step, which ranges from considering only atomic tactics to human-level proof steps (*i.e.*, compound tactics). Currently, our tool provides access to atomic tactics and as well as the capability to treat sequences of atomic tactics as a single proof step.

Second, tactic prediction may additionally require the synthesis of an argument. Some arguments to tactics such as `induction` or `rewrite` require the user to select an identifier in the local or global context to apply—what to do induction on and what to equality to rewrite by respectively. This can be considered a premise selection problem. Other arguments to tactics include synthesizing entire Coq terms. For example, the tactic `have: x := M` declares a local lemma that asserts that `M` is true and to introduce it into the context as `x` after it has been proven. Our tool provides the tactics, the arguments, and extracts local and global identifiers referenced in the arguments for convenience. This decomposes the problem of synthesizing tactic arguments into (1) predicting the identifiers involved and (2) constructing a term given a set of identifiers. The problem of synthesizing a term is difficult and a topic of research in itself—we do not address it in this paper.

## 5    REPRESENTING PROOF STATES

As we have just seen, both position evaluation and tactic prediction require a representation of proof states. Hence, we now discuss the representation of proof states in a form amenable for learning.

One manner in which the structured representation of proofs states provided by GamePad can be leveraged is to apply recurrent neural networks (RNNs) in a similar manner to how they are applied to parse trees in natural language processing to obtain an embedding vector. We can embed terms using their structure with a recursive embedding function $\mathcal{E} : \texttt{Term} \to \mathbb{R}^D$. For example, the embedding of an application term $M_0 \, M_1 \ldots M_r$ is obtained recursively as

$$\mathcal{E}(M_0 \ldots M_r) = \text{RNN}(\mathcal{E}'[\![\texttt{App}]\!], \mathcal{E}(M_0), \ldots, \mathcal{E}(M_r))$$

where $\mathcal{E}'[\![\cdot]\!]$ is a learnable embedding table indexed by the kind of the AST node (*e.g.*, `App` for application node). The leaves of the AST consist of constants, inductive types, constructors, existentials, and variables. Each constant, inductive type, constructor, and existential is given an entry in a learnable embedding table, which encodes the global state.

**Towards interpreter-inspired embeddings**   One way to add inductive bias to the embedding is to use the known reduction semantics of terms. For instance, a programming language interpreter uses an *environment*, a lookup table mapping variables to their values, so that the meaning of a variable is the meaning that the environment assigns it. For example, the meaning of the program expression `x` under an empty environment is a run-time error whereas the meaning of the same expression under the environment $\{\texttt{x} \mapsto 42\}$ is 42. We can apply this idea to obtain a new embedding function $\mathcal{E} : \texttt{Term} \times \texttt{Env} \to \mathbb{R}^D$ that additionally takes an environment $\rho$ of type `Env` which is a lookup table mapping variables to embedding vectors. Whenever we encounter a binding form such as a dependent product $\Pi x : M_1. M_2$ (similar to an anonymous function $\lambda x : M_1. M_2$), we bind a new meaning for $x$ within its scope (*i.e.*, the term $M_2$). To do so, we sample a random vector $v \sim \mathcal{N}^D$ according to a standard (multivariate) normal distribution[2] $\mathcal{N}^D$ and extend the local environment $\rho$ with the mapping $x \mapsto v$, written $\rho[x \mapsto v]$, so that mentions of the variable $x$ in $M_2$ can be looked up. Then the embedding of a binding form, such as a dependent product term (`Prod`), is obtained recursively as

$$\mathcal{E}(\Pi x : M_1. M_2, \rho) = \text{RNN}(\mathcal{E}'[\![\texttt{Prod}]\!], \mathcal{E}(M_1, \rho), \mathcal{E}(M_2, \rho[x \mapsto v])) \quad \text{where } v \sim \mathcal{N}^D .$$

The embedding for the variable case (which occur at the leaves of the AST) is $\mathcal{E}(x, \rho) = \rho(x)$, which corresponds to a variable lookup from the environment.

We resample the corresponding $v$ from $\mathcal{N}^D$ every forward pass in training when we embed the term $\Pi x : M_1. M_2$. By doing so, we encode the semantics that $x$ is just a placeholder and we should get the same result if we had used a different vector $v$ to embed it, while also preserving the environment lookup semantics that the embedding $v$ for $x$ is constant within the scope of $x$ in a single pass. Note

---

[2]We conjecture that we can use the type $M_1$ to put a better prior on the vectors sampled.

that the embedding is invariant by construction to variable renaming. Wang et al. (2017) propose another structured approach based on a De Bruijn term representation that is invariant under variable renaming that would be interesting to compare against. It would also be interesting to extend the entire embedding to more closely follow the structure of an interpreter so that it better reflects the *semantics* as opposed to the *syntax*, although we leave these extensions to future work.

**Embedding proof states**   After obtaining the embedding for each type[3] in the proof state and the embedding for the goal, we use another RNN over the embeddings. Note that the proof state context must be traversed in-order because the types of items later in the context may depend on the types of items that appear earlier in the context. As expected, the result of embedding a proof state is a vector in $\mathbb{R}^D$.

## 6   END-TO-END PROOF GENERATION FOR ALGEBRAIC REWRITES

In this section, we walk through a basic setup that uses GamePad to learn a simple algebraic rewriter. First, we use a deterministic procedure to synthesize Coq proofs for our domain and use GamePad to extract the resulting proof trees. Second, we train a tactic predictor and then deploy it to synthesize end-to-end proofs using GamePad's interactive mechanisms. This setup applies to any other domain of interest, provided we have a method of generating Coq proof scripts for that domain.

### 6.1   SIMPLE ALGEBRAIC REWRITE PROBLEM

We consider a problem that involves showing that two algebraic expressions are equivalent to one another. More concretely, we consider statements of the form:

$$\forall b \in G, X = b,$$

where $X$ is an arbitrary expression from the grammar $X ::= b \mid e \mid m \mid X \oplus X$ composed of elements with left-identity $e$, right-identity $m$, and binary operator $\oplus$. We have two simplification rules: $\forall b \in G, b \oplus m = b$ (right identity) and $\forall b \in G, e \oplus b = b$ (left identity).

Although the problem is simple, it involves bits of non-trivial reasoning. For instance, consider showing the equivalence $\forall b \in G, b \oplus (e \oplus m) = b$. Here, we can choose to eliminate the $e$ (left identity) or the $m$ (right identity). Notably, choosing to eliminate $m$ does not progress the proof because we cannot simplify the proof state $b \oplus e = b$. Note that the proof is not stuck because we can expand $b \oplus e$ back to $b \oplus (e \oplus m)$ and then get rid of $e$ the second time around. Thus, a prover has at least two choices in solving such problems: (1) maintain a global perspective to choose which parts of the goal to rewrite or (2) learn to expand terms. For this problem, we write a deterministic procedure that generates proofs of the first form that selects a position and an identity law, and attempt to learn this algorithm. We do not generate proofs that require backtracking (*e.g.*, due to a greedy rewrite) although it would be an interesting direction of future work.

### 6.2   END-TO-END PROOF SYNTHESIS

**Tactic prediction**   The tactic prediction model embeds a proof state into $\mathbb{R}^D$ and uses a fully-connected layer for prediction. We can model the proofs for this problem as a tactic prediction problem where the predicted category is a pair of the position in the AST and the identity to apply. We convert the position in the AST into a number using a preorder traversal of the Coq AST. For example, the second $\oplus$ in the expression $b \oplus (e \oplus m)$ has position 2. We can encode each identity as a single number. We obtain the prediction class as the pair of both numbers.

We implement end-to-end proof synthesis by combining a tactic prediction model that is trained offline with GamePad's lightweight interaction module. The interaction module takes care of reading in the current Coq proof state and sending tactic calls to advance the proof. We use the trained model to do inference on the current Coq proof state to obtain a position to rewrite and the identity law to apply. We then translate this to a tactic and take the corresponding action in Coq. For the current problem, we do not consider backtracking.

---

[3]Recall that Coq uses the same language to encode types and terms.

**Results** For this problem, we generate $400$ unique theorems of the form $\forall b \in G, X = b$ and their proofs where $X$ is a randomly generated algebraic expression of length $10$ that evaluates to $b$. We construct $X$ by recursively expanding the left and right expressions of $\oplus$ subject to the constraint that only one side, chosen at random, reduces to $b$ and the other side reduces to the appropriate identity (left or right). We stop when the length of the expression is $10$. We then extract the proof states using GamePad and train the tactic prediction model.

To test the model, we generate a distinct set of $50$ randomly generated algebraic expressions of length $10$ and test how many proofs our model can complete using a greedy approach. That is, at each proof state, we use the trained model to perform inference and choose the (rewrite position, identity law) tuple with the highest probability as our action. We find that such an approach can generate $14$ complete proofs, where we score a proof as a failure if any proof step fails. For expressions of length $10$, there are $9$ proof steps. We can relax this setting so that when any single proof step fails, we use the deterministic procedure to supply a proof step. This approach completes all $50$ proofs with an average failure rate of $1$ proof step per a proof.

We have observed cases where a good position is selected but the wrong identity law is paired with it. As we predict the position and rewrite jointly, this behavior is somewhat surprising. We also observe that the accuracy on the same test set for the tactic prediction task is $95\%$. Thus, while the imitation learning model has good generalization in the traditional sense, more work needs to be done to leverage the policy when synthesizing complete proofs.

# 7 REAL-WORLD DATA SETS

We can also apply GamePad to data extracted from real-world formalizations. In this section, we apply baseline position evaluation and tactic prediction models to data extracted from the Feit-Thompson formalization. We encounter difficulties not present in the simple algebraic rewrite domain here, including scaling and a more difficult tactic prediction problem.

**The Feit-Thompson data set** The Feit-Thompson theorem states that every finite group of odd-order is solvable, and is a deep result in group theory. The formalization has the interesting property that the researchers attempted to follow the book proofs as closely as possible. The extracted data set consists of $1602$ lemmas and expands into $83478$ proof states. For our tasks, we split the lemmas in the data set into a training, validation and test set in the ratio $8 : 1 : 1$, and ensure that the number of proof states in the splits are in a similar ratio. Note that we split by lemmas and not by proof states because predictions for proof states within the proof of the same lemma are not independent and can lead to a more optimistic evaluation of the generalization capability of the models (particularly for the case of position evaluation).

As a reminder, each proof state consists of a context and an associated goal. Each context contains on average $60$ terms or identifiers, and on average $4000$ nodes at the kernel level (and $1000$ at the mid level without implicit arguments). The most common nodes include constants, applications, and variables, and as such, we focus on those during the design of our embedding. A formalization such as CompCert would contain a different distribution of AST nodes. In particular, as it concerns program verification, there would be more AST nodes involving fixed-points[4], case analysis, and inductive type constructors.

The most prevalent tactics include `rewrite` and `have`. As a reminder, a `rewrite` tactic requires the user to indicate which equality in the current local or global context to apply and is akin to premise selection. The `have` tactic introduces an intermediate lemma in the proof and is the hardest to learn as the user usually uses some mathematical insight about the problem before conjecturing a statement. We currently do not synthesize arguments for `have` tactics, although our tool extracts such data. We believe a generative model for synthesizing them would be a great direction for future work.

---

[4]One difficulty with embedding fixed-points is that we need the embedding of the body in order to embed the body. An interesting approach here is to introduce a loss $|\mathcal{E}(f(x)) - \mathcal{E}(x)|^2$, where we start with learnable random embeddings for the base cases.

Table 1: Training time speedups for the GRU model with state size of 128 on the position evaluation task obtained compared to an un-optimized baseline. [*] indicates CPU and [†] indicates GPU.

| Model | Base[*] | Embedding Sharing[*] | Dynamic Batching[*] | Both[*] | Both[†] |
|---|---|---|---|---|---|
| Speedup (approx.) | 1 | $10\times$ | $10\times$ | $130\times$ | $190\times$ |

Table 2: Test accuracies for position evaluation (Pos) and tactic prediction (Tac). † indicates kernel-level. ‡ indicates mid-level without implicit arguments. For tactic argument prediction, we report validation recall for models with a minimum precision of $10\%$

| Model | Pos[†] | Pos[‡] | Tac[†] | Tac[‡] | Tac[†] arguments |
|---|---|---|---|---|---|
| Constant | 53.66 | 53.66 | 44.75 | 44.75 | - |
| SVM | 57.37 | 57.52 | 48.94 | 49.45 | - |
| GRU | 65.30 | 65.74 | 58.23 | 57.70 | 25.98 |
| TreeLSTM | 68.44 | 66.30 | 60.63 | 60.55 | 23.91 |

**Scaling to large proof trees** In practice, ASTs can be on the order of thousands of nodes. We can apply two optimizations to scale to larger trees. The first optimization involves *embedding sharing*, where we memoize the embedding and the associated computation graph for any expression or subtree when it appears another time in the same proof state. For instance, if the context has terms $M_1 \, M_2$ and $M_1$, the embedding for $M_1$ is computed once. A single forward and backward pass is thus performed on this computation graph, with the gradients automatically accumulating from all places where the embedding was used. It thus helps save both memory and computation time of our models. The second optimization involves *dynamic batching* (Looks et al., 2017; Polosukhin & Zavershynskyi, 2018), which enables us to efficiently batch the computation of ops that perform the same operation albeit on different inputs, and thus make better use of the parallel acceleration provided by multi-core CPU's and GPU's. [5]

Table 1 shows the approximate speedups obtained from applying embedding sharing and dynamic batching to representing proof states. Note that the GPU speedup will increase with larger models.

## 7.1 POSITION EVALUATION AND TACTIC PREDICTION

We use the interpreter-inspired embeddings to embed proof states into $\mathbb{R}^D$, and then aim to train models for position evaluation and tactic prediction tasks. For the position evaluation task, we binned the target predictions into $K = 3$ classes—close ($< 5$ steps), medium (between $6 - 19$ steps, inclusive), and far ($> 20$ steps), and perform a simple three way classification.

The tactic prediction is more complicated. First, we group tactics into equivalence classes if they perform the same mathematical operation. For example, we group the tactics `reflexivity` and `done` together because they are applied at the end of proofs to show that a statement is trivially true. Second, we now also have tactic arguments. We train two models, one that predicts only the tactic and another that additionally predicts the arguments. The first is a 23 way classification problem. For the second, the arguments could be (1) a term from the local context, (2) a term from the global context (*e.g.*, a lemma), or (3) a term created by the human user. As arguments in the third category can be any Coq term, the space of arguments is potentially infinite. As a reminder, the data set makes extensive use of `have` tactics—in essence, this would require the model to conjecture a statement. For our baseline models, we focus on arguments in the first category and predict the presence or absence of each term in the context at any position in the arguments. For each term, we use the final hidden state and the embedding of each term followed by a linear layer to produce a two way prediction. Note that the distribution of labels is skewed towards the absent category. Thus, we are more interested in the precision-recall curve. We weigh the cross-entropy loss higher for presence class, and also try to balance the distribution of labels by randomly sampling only a subset of the negative class at training time.

---

[5]It is also possible to share embeddings across mini-batches, but the trade-off is that we will be using stale parameters. One can potentially further speed up training by a factor of about two by moving the creation of the computation graph to a separate thread

**Results**   We first start by training a simple SVM (Cortes & Vapnik, 1995) on a set of heuristic features like context size, goal size, number of hypothesis in context and the smallest edit distance between a hypothesis and the context. The SVM performs better than the constant baseline of guessing the most common class. We then train RNN models to utilise our embedding strategy. We train GRU (Cho et al., 2014) and TreeLSTM (Tai et al., 2015) models using mini-batches of 32 proof states, set the RNN state size to 128, and use the Adam optimizer Kingma & Ba (2015) with a learning rate of 0.001. We use input (Srivastava et al., 2014) and weight (Merity et al., 2017) dropout with a rate of 0.1 for the TreeLSTM models (higher rates led to much slower learning). All neural net models were trained using PyTorch (Paszke et al., 2017). Table 2 show the results for the tasks. We were able to improve upon the SVM baseline, which indicates that it is possible to learn useful representations using the human supervision data and utilizing our proof state embeddings. We then experiment with removing the bookkeeping aspects of the prover by switching from kernel level to mid level proof states without implicit arguments. We obtain similar accuracies, indicating that most of that data is redundant.

## 8   RELATED WORK

The *level of abstraction* and *representation* of proofs that learning is applied to are salient points of comparison between work on learning and theorem proving. These choices inform the setup and challenges associated with the learning problem.

As in our work, there are systems that experiment with learning in ITPs. Duncan (2002) explores how to learn (user-defined) tactics from a corpus of proofs in Isabelle so that they can be applied to future proofs. ML4PG (Komendantskaya et al., 2012) interfaces to ITPs at the level of the user interface for entering proof scripts. Thus, ML4PG is applicable to multiple ITPs although it obtains less granular proof states. Holphrasm (Whalen, 2016) uses string encodings of proof states and focuses on tactic argument synthesis (there is essentially only one tactic in the underlying ITP MetaMath (Megill, 2007)). HolStep (Kaliszyk et al., 2017) addresses premise selection using string-level encodings of proof states. Wang et al. (2017) extend the HolStep work to show the advantage of using a DeBruijn representation of proof terms as opposed to string-level encodings for premise selection. The structured representation provided by GamePad would support experimenting with such extensions. Gauthier et al. (2017) explores learning tactic-level proof search in Isabelle using hand-crafted features on string encodings of proof states. It would be interesting to experiment with their algorithm to our algebraic rewrite problem. Nagashima & He (2018) looks at explainable tactic prediction in Isabelle.

Other approaches focus on automated theorem provers, which are designed to prove theorems with no human interaction. Irving et al. (2016) describes the premise selection problem and trains neural network models on proof traces obtained from applying E (Schulz, 2002) to the Mizar Mathematical Corpus. Loos et al. (2017), in addition to addressing premise selection, also address a *clause selection* task by applying neural network models to this problem in E. Kaliszyk & Urban (2014) demonstrate that similar learning based methods can prove 39% of the lemmas in the Flyspeck project (Kaliszyk & Urban, 2014).

Another take on learning and theorem proving is to replace an entire theorem proving (sub)routine with a learned algorithm instead of using learning for heuristics. For instance, end-to-end differentiable proving (Rocktäschel & Riedel, 2017) replaces traditional *unification* with a trained neural network and demonstrates the efficacy of this approach for knowledge base completion. Neurosat (Selsam et al., 2018) applies a neural network model to predict *satisfaction problems* and shows how to recover a satisfying assignment.

## 9   CONCLUSION

In this work, we look at theorem proving problem through the lens of a system that enables learning with proofs constructed with human supervision. We highlight three key aspects of the problem at this level. The first concerns obtaining inputs to a learning algorithm that approximate the level of abstraction faced by a human prover. For this, we use an ITP, as it retains aspects of human supervision. GamePad preserves the structure of the proofs (*e.g.*, annotations regarding implicit arguments) so they can be used for building models. The second involves building models that

employ the game-like structure of ITP proofs. Here, we experiment with tactic prediction for toy and real world data sets. Finally, as a consequence of theorem proving at a higher-level (compared to SMT solvers), we will need to be careful to distinguish the syntax from the semantics of terms. Our current approach is to provide structured representations of terms so that more semantic structure can be exploited. While our results are preliminary, our hope is that GamePad provides an accessible starting point to explore the application of machine learning in the context of interactive theorem proving.

## 10 Future Work

We end by suggesting a few avenues for extending our work. The first concerns the design of new benchmarks for human-level proofs. In this paper, we designed a relatively simple algebraic rewrite problem to test the system end-to-end. Designing more difficult problems that still admit tractable learning (such as solving infinite sums or integrals) would be a great direction for future work. Note that you can define a new domain inside Coq and use GamePad to build provers that learn from example proofs. A second direction concerns building models that conjecture and explore the space of true statements, in addition to proving statements. This is particularly important in synthesizing arguments to tactics like `have`. Currently, we only predict the tactic identifiers and do not synthesize the entire term. Building a generative model would be a great next step. Lastly, it would be interesting to see if using end-to-end training with reinforcement learning and utilizing Monte-Carlo tree search to efficiently explore the search space can be effectively applied to human-level proofs.[6]

## 11 Acknowledgements

We would like to thank Diederik Kingma, Tim Salimans, and Geoffrey Irving for reviewing initial drafts of the work. We thank Daniel Selsam for suggesting that we consider removing implicit arguments, and Jonathan Cai for discussions about the toy problem. Daniel Huang was supported by DARPA FA8750-17-2-0091.

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
