# OpenReview forum: "GamePad: A Learning Environment for Theorem Proving"
_ICLR.cc/2019/Conference_

### Official Review · AnonReviewer1 · 2018-11-02
**An intriguing integration of ML and automated theorem proving**

**Rating:** 7
**Confidence:** 2

**Review:**

Summary: This paper mixes automated theorem proving with machine learning models. The final goal, of course, is to be able to train a model that works in conjunction with an automated theorem proving system to efficiently prove theorems, and, ideally, in a way that resembles the way humans prove theorems. This is a distant goal, and the authors instead focus on several tractable tasks that are required for future progress in this direction. They start by integrating the Coq theorem proving environment with ML frameworks, allowing for the creation of models that perform various tasks related to theorem proving. In particular, they focus on two tasks. One is to estimate how many steps are left to complete the proof given a current proof state. The other is to determine what is a good choice of next step. Finally, they also consider issues surrounding representations of the various data structures involved in proofs (i.e., the proof tree, variables, etc.). They test various models on a synthetic nearly trivial logical expression proof, along with a more complicated (and meaningful real world) group theory result.

Strengths: This is a very important area. Automated theorem proving has a potentially very significant impact, and being able to take advantage of some of the recent successes in ML would be excellent. The main environment proposed here, integrating PyTorch with Coq could potentially be a very useful platform for future research in this area. The paper exposes many interesting questions, and I generally think we need more exploratory papers that open up an area (as opposed to seeking to finalize existing areas)

Weaknesses: The paper is pretty tough to understand without a lot of background in all of the existing theorem proving work (which might be fine for a conference in this area, but for this venue it would be nice to be more self-contained). The organization could also use some work, since it's often tough to figure out what the authors actually did. The experimental results seem very preliminary---although it's hard to say, as there is no easy way to compare the results to anything else out there. In general a lot of details seem missing.

Verdict: The authors admit this is a preliminary work, and I agree with that. The paper certainly introduces many more questions than it answers. However, I think that in this case it's a good thing, and this type of paper has the potential to inspire a lot of new and exciting research, so I voted for acceptance.

Comments and questions:

- As mentioned, a lot of the terminology is introduced very quickly and could stand to be more self-contained, i.e., "tactics" could be defined as being simple transformations that are applied to a current proof state to obtain another proof state, and each language has a library of tactics available.

- Probably the major contribution of the work is the integration of the CoQ and Pytorch, so a bit more content describing how the Python data structures that wrap around Coq structures would be interesting here.

- I didn't really understand one of the major contributions: the embedding function for the M_i conditioned on the environment. How does the sampled Gaussian vector work here? In general this section is pretty confusing, it would be great to include a schematic to show how the different levels of embeddings for different structures work here.

- How does the real-world dataset work? Does the dataset contain one automated proof of the entire theorem, or several different proofs (ultimately produced by different user choices)? Are you measuring accuracy on the proofs of individual lemmas?

---

> ### Author Response · Authors · 2018-11-26
> **Re: An intriguing integration of ML and automated theorem proving**
>
> - I didn't really understand one of the major contributions: the embedding function for the M_i conditioned on the environment. How does the sampled Gaussian vector work here? In general this section is pretty confusing, it would be great to include a schematic to show how the different levels of embeddings for different structures work here.
> Thanks for pointing it out, we’ve clarified this section in the paper (see our main comment above).
>
> - How does the real-world dataset work? Does the dataset contain one automated proof of the entire theorem, or several different proofs (ultimately produced by different user choices)? Are you measuring accuracy on the proofs of individual lemmas?
> Proving the final theorem involves first proving other theorems, which are then used as lemmas in the proof of the final theorem. The theorems were proved by a team of researchers following the SSReflect style of structuring proofs, and each theorem has one proof. The dataset includes all these theorems, and we extract the individual proof steps from each of these for the position evaluation and tactic prediction tasks.

---

### Official Review · AnonReviewer2 · 2018-11-03
**Nice exposition on the opportunities and the issues in using machine learning for interactive theorem proving in Coq**

**Rating:** 7
**Confidence:** 3

**Review:**

In the paper, the authors describe how machine learning techniques can be used to help build proofs in the widely-used interactive theorem prover Coq. They do so by explaining the experience with their system called GamePad, which converts various proof-related objects of Coq to python data structures so that python-based machine learning tools can be applied to those data structures.

Although the word, GamePad, appears in the title of the paper, the paper focuses mostly on how Coq works, which aspects of proving in Coq can be aided by machine learning techniques, and what challenges they experienced when using machine learning techniques to the Feit-Thompson data set, an impressive big Coq proof of a famous theorem in group theory. I wasn't impressed by the GamePad tool, which seems to be just a translator of Coq internal data structures to python data structures. But I liked the authors' general exposition about the opportunities and the issues in using machine learning techniques to theorem proving in Coq. They explain that one key difficulty of the tactic prediction problem is the need to synthesize a term parameter to a tactic. They also point out the issue of choosing the granularity of tactic when approaching this problem.

I give positive score mainly because some other audience in ICLR may learn about a new problem domain and get excited about it by interacting with the authors of the paper.

Here are some minor comments.

* Caption of Figure 1: its goal that the statement ===> its goal the statement

* p3: function K ===> function M

* p5: In the interpreter-inspired embedding of a dependent product, are you drawing a real number from a normal distribution for every occurrence of v in the proof once, and using the drawn real number for the occurrence whenever the occurrence is referred to later? Or when it is referred to later, are you drawing a new real number?

* p6: How did you generate the training data for the experiment reported in Section 6?

---

> ### Author Response · Authors · 2018-11-26
> **Re: Nice exposition on the opportunities and the issues in using machine learning for interactive theorem proving in Coq**
>
> - p5: In the interpreter-inspired embedding of a dependent product, are you drawing a real number from a normal distribution for every occurrence of v in the proof once, and using the drawn real number for the occurrence whenever the occurrence is referred to later? Or when it is referred to later, are you drawing a new real number?
> Thanks for pointing it out, we’ve clarified this section in the paper (see our main comment above).
>
> - p6: How did you generate the training data for the experiment reported in Section 6?
> For this problem, we generate $400$ unique theorems of the form $\forall b \in G\ldotp X = b$ and their proofs where $X$ is a randomly generated algebraic expression of length $10$ that evaluates to $b$. We construct $X$ by recursively expanding the left and right expressions of $\oplus$ subject to the constraint that only one side, chosen at random, reduces to $b$ and the other side reduces to the appropriate identity (left or right). We stop when the length of the expression is $10$. We have updated the paper with these details.

---

### Official Review · AnonReviewer3 · 2018-11-04
**Interesting direction, but too preliminary**

**Rating:** 4
**Confidence:** 4

**Review:**

The submission describes a system for applying machine learning to interactive theorem proving. The paper focuses on two tasks: tactic prediction (e.g. attempting a proof by induction) and position evaluation (the number of  remaining steps required for a proof). Experiments show that a neural model outperforms an SVM on both tasks, using proof states sampled from a proof of the Feit-Thompson theorem as a dataset. It's great to see work on applying neural networks to symbolic reasoning. The paper is clearly written, and provides helpful background on interactive theorem proving.

The main weakness of the paper is the limited experiments, which only really show that neural methods outperform an SVM (with only a high level description of the features) - and only on the proof of a single theorem. The paper doesn't explore relevant interesting questions, such as whether the model is helpful for guiding humans or machines in making proofs, or perhaps if the approach can be used to find more human-understandable proofs than those found without training on human data. What are the trade-offs in learning from human proofs instead of automated proofs?

Overall, the paper explores an interesting direction, but I think the current experiments are too preliminary for acceptance.

---

> ### Author Response · Authors · 2018-11-26
> **Re: Interesting direction, but too preliminary**
>
> -The main weakness of the paper is the limited experiments, which only really show that neural methods outperform an SVM (with only a high level description of the features) - and only on the proof of a single theorem.
> We note that the Feit-Thompson dataset consists of 1602 theorems proved by a team of researchers over 6 years. The paper introduces a tool for applying ML to theorem proving in Coq, describes tasks and challenges that are relevant for tactic-based ITP’s, creates appropriate datasets for those tasks; and thus the aim of the experiments, while preliminary, was to provide baselines / introductory experiments for our framework and concretely explore some of the issues with applying ML to tactic-based ITP.
>
> -The paper doesn't explore relevant interesting questions, such as whether the model is helpful for guiding humans or machines in making proofs, or perhaps if the approach can be used to find more human-understandable proofs than those found without training on human data. What are the trade-offs in learning from human proofs instead of automated proofs?
> Thanks for the suggestions. We experimented with end-to-end proving in a simple algebraic rewrite setting to test how well the tactic prediction model works for that problem. Addressing end-to-end proving in real world formalizations was beyond the scope of our project due to the difficulty of synthesizing existentials.

---

> ### Comment · Area_Chair1 · 2018-11-30
> **Please consider other reviews**
>
> Reviewer 3,
>
> Thank you for your review. Your score is currently the outlier amongst the reviews, which is fine! However, given the substance of the other reviews, the author response below, and the revisions made to the paper, it would be good to get a bit more detail from you. Could you please read the author response and other reviews, as well as revisions made to the paper, and either consider revising your assessment or providing an explanation as to why you stand by your score?
>
> There are a few days left for you to engage in discussion with the authors and fellow reviewers, so you may wish to do that first.

---

### Author Response · Authors · 2018-11-26
**Thanks to reviewers for feedback, uploaded revised draft**

We would like to thank all reviewers for their reviews and their constructive feedback. We address reviewer-specific questions individually. We have uploaded a revised draft incorporating their feedback. Specifically:
- Fixed typos identified by reviewer 2.
- Thanks to the reviewers for pointing out that the embedding section was confusing. There was a typo in the formula that described the interpreter embedding. The rhs formula was supposed to represent how the environment $\rho$ changed when we recursively embed $M_2$ during the embedding of $\Pi x: M_1. M_2$.  We have since modified it to represent the complete embedding of the term $\Pi x: M_1. M_2$. We have also clarified the section in the paper, summarized below. For each binding form such as $\Pi x: M_1. M_2$, the Gaussian vector $v$ representing $x$ is kept fixed in the scope of $x$, i.e., wherever $x$ appears in $M_2$, in a single forward pass so that it reflects the environment lookup semantics. However, it’s resampled for each forward pass so that it encodes the semantics that $x$ is simply a placeholder and we should get the same result if we had used a different vector $v$ to embed it.
- We explained how the expressions in the simple rewrite problem are randomly generated and clarified the form of the statements proved in the simple rewrite section with its grammar.

---

### Public Comment · ~reza_haghi1 · 2019-07-01
**gamepad**

hi...
i think Sony DualShock 4 Controller is very amazing...
dualshock 4 is one of the best computer gamepad in the world... i realy like this

please check my website for this:
https://www.plaza.ir/mag/139803/148890/best-gamepads-2019/

---

### Meta-Review · Area_Chair1 · 2018-12-13
**A great starting point for ML-assisted ITP**

**Confidence:** 4
**Recommendation:** Accept (Poster)

**Metareview:**

This paper provides an RL environment defined over Coq, allowing for RL agents and other such systems to to be trained to propose tactics during the running of an ITP. I really like this general line of work, and the reviewers broadly speaking did as well. The one holdout is reviewer 3, who raises important concerns about the need for further evaluation. I understand and appreciate their points, and I think the authors should be careful to incorporate their feedback not only in final revisions to the paper, but in deciding what follow-on work to focus on. Nonetheless, and with all due respect to reviewer 3, who provided a review of acceptable quality, I am unsure the substance of their review merits a score as low as they have given. Considering the support the other reviews offer for the paper, I recommend acceptance for what the majority of reviewers believes is a good first step towards one day proving substantial new theorems using ITP-ML hybrids.